# Psychological Capital and Turnover Intention: The Mediating Role of Burnout among Healthcare Professionals

**DOI:** 10.3390/ijerph21020185

**Published:** 2024-02-06

**Authors:** Laura Zambrano-Chumo, Ruben Guevara

**Affiliations:** CENTRUM Catolica Graduate Business School, Pontificia Universidad Catolica del Peru, Lima 15023, Peru; rguevara@pucp.pe

**Keywords:** psychological capital, burnout, turnover intention, healthworkers, positive psychology

## Abstract

Psychological capital (PsyCap) has been identified as an individual’s positive psychological state of development that can help to reduce negative organizational outcomes. However, there is still a gap in understanding how PsyCap influences different aspects of organizations. This study investigates the mediating role of burnout in the relationship between PsyCap and turnover intentions among healthcare professionals. A cross-sectional survey was conducted among 320 healthcare professionals. The estimation of the relationships between PsyCap, burnout, and turnover intentions was obtained through structural equation modelling (SEM). A mediation analysis was carried out using bootstrapping procedures. The results show that burnout has a significant and positive effect on turnover intention and PsyCap has a significant and negative effect on burnout. Moreover, burnout fully mediates the relationship between PsyCap and turnover intention. These findings suggest that PsyCap can effectively reduce negative outcomes like burnout, but its positive impact may be limited when considering other outcomes such as turnover intention.

## 1. Introduction

Turnover intention, the desire of employees to voluntarily leave their jobs, is increasingly concerning, especially in the healthcare sector where it leads to higher payroll costs and necessitates proactive intervention. High turnover rates, linked to employee burnout and reduced mental well-being, have significantly impacted healthcare over the last decade. This trend not only affects employee retention but also hinders organizational competitiveness due to the loss of qualified personnel [1,2,3,4,5,6,7,8].

Research grounded in positive psychology emphasizes the importance of psychological capital, showing its correlation with various employee attitudes, behaviors, and performance outcomes [9,10,11]. Key findings reveal that psychological capital positively affects crucial aspects like employee engagement, job satisfaction, and overall job performance. Further studies support that individuals with higher psychological capital are better at managing job-related stress, which in turn reduces their intention to leave their positions [12,13,14].

A study focusing on the mediating role of psychological capital suggests interventions to strengthen it could reduce burnout [15]. Although its direct impact on turnover intentions is statistically insignificant, its significant indirect effect through occupational stress is notable [13]. These results highlight the crucial role of positive psychological capital in influencing employee well-being, attitudes, and intentions, suggesting the need for interventions aimed at improving mental health and reducing turnover intention.

The role of negative mental states like burnout is crucial in understanding turnover intention [16]. Burnout, often seen in professionals with high-stress interactions, leads to competence issues, negative emotions, and a sense of ineffectiveness [14,17]. Healthcare professionals face the highest burnout rates due to stressful demands and patient care challenges [18,19,20,21]. This has significant implications for patient safety and turnover rates, marking it as a global issue in healthcare [22]. Burnout is increasingly common, necessitating further research, especially in healthcare personnel in developing countries where work conditions differ markedly from developed countries [23,24].

Employee turnover introduces uncertainty in the workforce and economic losses due to recruitment, selection, and training costs [1,2]. Researchers are exploring how high-performance work practices influence employee behavior, particularly turnover intention, in healthcare professionals in developing countries [25]. Understanding the decision-making process behind their professional disengagement is crucial. Burnout, a common syndrome in healthcare institutions [19], has been exacerbated by the COVID-19 pandemic, leading to negative psychosocial and physical impacts on healthcare personnel [26]. Improving their psychosocial well-being is essential to reduce turnover rates and enhance the quality of services, especially under the increased pressure of the pandemic [27].

Starting a new job is often a stressful experience for employees, who have to adapt to new tasks and expectations [28]. Studies have shown that high levels of psychological capital can reduce negative behaviors like turnover intention [29], with a significant negative relationship observed between positive psychological capital and turnover intention [30]. However, while employee turnover is harmful to both individuals and organizations, the role of psychological capital in mitigating this issue, particularly in developing countries, needs further exploration [31]. Studies on turnover intention have primarily focused on job-related characteristics, often overlooking factors like burnout [32]. The relationship between job-related stress, psychological capital, and turnover intention is complex, with some studies showing contradictory results, especially regarding the influence of psychological capital [29,33]. These inconclusive findings, varying across different contexts, highlight the need for more research in developing countries to better understand these dynamics.

The World Health Organization (WHO) emphasizes the need for research on burnout [34], a condition particularly relevant in professions requiring intense interpersonal interactions [35,36]. The COVID-19 pandemic has further highlighted the importance of addressing healthcare workers’ mental health beyond mental disorders. Research shows that depressive symptoms like burnout affect turnover intention [37], leading to increased absences and reduced work efficiency in healthcare. Addressing burnout is essential to maintain productivity, morale, and health of employees [38]. In nursing, burnout significantly impacts turnover intention [39], and studies suggest that positive psychological capital may mitigate its effects, enhancing job performance even in burnout situations. Positive psychology and psychological capital, which represent a positive mental state of personal enhancement, have been shown to improve burnout and turnover intention levels [40,41,42]. However, the diverse roles of psychological capital—as an exogenous, moderating, or mediating variable—yield varied results, indicating the need for continued exploration and discussion of its theoretical and practical implications [43].

The inconclusive findings regarding the effects of psychological capital on turnover intention suggest the need for further research on this relationship. This priority is substantiated by the issue of high job-related stress and turnover intention, which has led to increased healthcare problems and management costs [13]. Finally, understanding how to reduce turnover intention and improve employee retention holds paramount significance for organizations [1]. Researchers suggest that for future studies, additional variables should be included, along with psychological capital and other related constructs, and a more in-depth exploration of their combined effect on job-related stress [13]. For this reason, research is needed to understand, from the theory of planned behavior, the theory of reasoned action, the decision-making mechanisms of healthcare personnel disengagement, and from positive psychology, how positive psychological capital can improve levels of burnout and turnover intention.

To fill this gap in the literature, this article aims to analyze the mediating role of burnout in the relationship between psychological capital and turnover intention among healthcare professionals.

### 1.1. Psychological Capital and Turnover Intention

Positive psychology centers on the affirmative dimensions of human behavior and is fundamentally concerned with enhancing happiness and overall well-being [44]. It also places significant emphasis on various facets of life, mental health prevention, and the promotion of life satisfaction [45]. Within the positive psychology framework, psychological capital, along with its constituent resources, is recognized as a constructive state [46,47,48,49]. Meeting the criteria of positive psychological constructs, these resources encompass hope, resilience, optimism, and self-efficacy, and their amalgamation is referred to as psychological capital [10,50,51].

Psychological capital is defined as a positive mental state of personal improvement, characterized by (a) having confidence (self-efficacy) to undertake and invest the necessary effort to thrive in challenging responsibilities; (b) making positive attributions (optimism) about success both now and in the future; (c) directing efforts toward goals and, when necessary, altering paths towards objectives (hope) with a specific end in mind for success; and (d) when confronted with problems and adversities, managing, enduring, and even overcoming them (resilience) to achieve success [41] (p. 2).

On the other hand, Porter and Steers concluded that future research in psychology should place greater emphasis on the decision-making process related to employee turnover. This process involves several intermediate steps leading to the ultimate decision to leave, a step taken following dissatisfaction during the process [52]. Shaw also asserted that individuals with high levels of positive affect tend to change jobs when dissatisfied with their current work [53]. Mobley further argued that job dissatisfaction translates into thoughts of leaving, with expectations of finding a more satisfying job. Subsequent studies have defined turnover intention as the subsequent withdrawal behavior when employees experience dissatisfaction [23].

The concept of turnover intention can be defined as the final step in the decision-making process before an individual actually leaves their job [54]. On the other hand, Tett and Meyer defined it as the conscious and deliberate willingness to leave the organization [55]. Drawing from Ajzen’s Theory of Planned Behavior and Fishbein and Ajzen’s Theory of Reasoned Action, turnover intention has allowed for the development of models that position intention as a significant predictor of actual behavior, i.e., the act of resigning from one’s job [56].

According to Bothma and Roodt, contextual factors such as labor market conditions and employability also determine turnover intention [57]. In other words, an individual’s intention to leave a job is related to their perception of their prospects of finding another one. Additionally, turnover intention has been examined in relation to various constructs, including job satisfaction [7,25,58,59,60], mental health and well-being, leadership [29], organizational commitment [25], social support [61], professional identity [62], organizational climate [3], organizational citizenship behavior [63], and others. Furthermore, there is a substantial body of research investigating the relationship between turnover intention and burnout [22,24,39,64,65,66,67,68,69,70], as well as its connection with the construct of psychological capital [25,58,71,72,73,74].

There is evidence indicating that enhancements in psychological capital and social support, along with a reduction in occupational burnout, play a pivotal role in diminishing turnover intention [75]. In another study, the results unveiled that psychological capital exerted a negative and statistically significant influence on turnover intention. Furthermore, psychological capital functioned as a partial mediator in the association between occupational stress and turnover intention [30]. The findings imply that occupational stress operates as a complete mediator between psychological capital and turnover intention and as a partial mediator between relational social capital and turnover intention. Notably, none of the three components of capital exhibited a direct and significant effect on turnover intention. Consequently, healthcare personnel management should prioritize the reinforcement of humanistic care, optimization of incentive mechanisms, and enhancement of internal institutional management to foster stability and reduce turnover rates [13].

Under this framework, a meta-analysis conducted by Avey and colleagues establishes a negative association between psychological capital and undesirable attitudes, such as turnover intention [3]. These findings are substantiated by another study, which identifies psychological capital as a crucial personal resource with a negative impact on turnover intention [31]. It is postulated that individuals possessing higher levels of psychological capital are better equipped to effectively navigate workplace challenges, consequently resulting in reduced turnover intention [13]. Additionally, a separate study concludes that a significant negative influence on turnover intention stems from positive psychological capital [33]. Building on these results that illustrate the inverse relationship between psychological capital and turnover intention, the following hypothesis was proposed:

**Hypothesis 1 (H1).** 
*Psychological capital has a significant and negative effect on turnover intention.*


### 1.2. Psychological Capital and Burnout

The concept of burnout was first introduced in the literature in 1974, suggesting that it is the result of prolonged workplace stress [36]. Burnout contributes to poor job performance, low productivity, high absenteeism, and employee turnover, and it can have a negative impact on colleagues [20]. Burnout is a syndrome involving emotional exhaustion and depersonalization, with reduced personal accomplishment, resulting in ongoing workplace stress that has not been effectively managed [16].

Burnout results from a lack of resources and the inability to meet job requirements or from an imbalance between individual effort and compensation [76]. The Job Demand-Resource Theory postulates that job demands lead to heightened physical and mental exertion, positively correlating with burnout. In contrast, job resources foster employee well-being. Therefore, psychological capital is considered an individual resource that exerts a similar influence as job resources, resulting in a negative association between the two [77].

Building upon these principles, previous studies have established a noteworthy indirect relationship between burnout and psychological capital [43,77,78,79,80,81]. Additionally, in one research study, it was discerned that psychological capital not only predicts burnout but also significantly influences it [42].

Researchers have posited that since positivity and negativity often coexist within any context, it is crucial to integrate and test these factors together rather than separately [8]. In formulating the following hypothesis, we take into consideration findings that emphasize the significant role of positive psychological capital. These studies reveal that this construct serves as an indirect mediator between burnout and job performance [15].

Additionally, other results have indicated moderate and negative correlations between work addiction and psychological capital, as well as between psychological capital and burnout. Moreover, in the context of a developed country, research conducted among nursing personnel has demonstrated that positive psychological capital serves as a predictive variable for burnout [42]. Given these scholarly suggestions for further exploration of the relationship between these two constructs, we propose to corroborate the following hypothesis.

**Hypothesis 2 (H2).** 
*Psychological capital has a significant and negative effect on burnout.*


### 1.3. Burnout and Turnover Intention

An extensive body of literature addresses how burnout affects turnover intention. There is evidence showing a significant and positive relationship between burnout and turnover intention in a study conducted with students [65]. A similar result was observed in a research study involving a sample of teachers in the United States [22] and a study with a sample of nurses [69]. The study by Chen and colleagues examined the relationship between turnover intention and burnout, finding a significant and positive association between these two constructs [23]. However, the results revealed that job satisfaction only exerts a mediating effect of low impact.

In a study aimed at analyzing the mediating role of job satisfaction between burnout and turnover intention, a positive relationship was reported between the cynicism dimension of burnout and turnover intention, while the exhaustion dimension was not significant [68]. Similar findings were also found in another study, which also confirmed the mediating role of job satisfaction between burnout and turnover intention among primary healthcare personnel [24]. Additionally, scholars found a positive effect between burnout dimensions, emotional exhaustion, depersonalization, and turnover intention [66]. Moreover, this study analyzed the moderating role of subjective vitality, characterized by a positive sense of vitality and energy. The authors identified a significant moderating role of these variables, meaning that the strength of the relationship between burnout dimensions and turnover intention increases when employees have high levels of subjective vitality.

Moreover, researchers found that occupational burnout is closely related to turnover intention and is positively influenced by both stress and role ambiguity. Furthermore, a statistically positive association was found between work–life imbalance and burnout [82].

Some findings have demonstrated that depressive symptoms, secondary traumatic stress, burnout, and compassion satisfaction have an impact on turnover intention [37]. An analysis of turnover intention data revealed a significant intention among nursing staff to completely leave their positions. Consequently, the correlations in these studies indicate a positive relationship between turnover intention and job burnout [20,83]. In a separate study, burnout was introduced as a mediating variable. The outcomes of this investigation clarified that burnout serves as a mediator in the association between workplace harassment and turnover intention [84]. Among nursing personnel, research has highlighted that burnout significantly influences turnover intention [39]. Building on these outcomes, another study also identified a positive correlation between burnout and turnover intention [23]. These findings support the formulation of the following hypothesis:

**Hypothesis 3 (H3).** 
*Burnout has a significant and positive effect on turnover intention.*


### 1.4. The Mediating Role of Burnout

Personnel turnover instigates workforce uncertainty and presents a significant concern for organizational management. This construct has gained prominence as a critical factor impacting an organization’s competitiveness. Hence, researchers advocate for its investigation in conjunction with other related variables, including job satisfaction, burnout, and psychological capital. The latter has garnered attention within the scientific community due to its recognized positive effects on organizational outcomes.

An illustrative example pertains to a study focused on elucidating the influence of psychological capital on both job stress and job turnover intentions. It was revealed that psychological capital exerts a significant negative influence on both variables. Furthermore, the research identified a noteworthy connection between job stress and employees’ intentions to quit their jobs. Likewise, a moderation analysis indicated that job stress functions as a mediating factor between psychological capital and turnover intentions [71]. These findings align with those of previous studies [10,11,73,75]. Other scholars have investigated the mediating role of psychological capital, demonstrating that it generally mediates the relationship between occupational stress and job burnout [78]. Consequently, psychological capital emerges as a potential positive resource for mitigating the adverse effects of occupational stress on job burnout and alleviating the experience of job burnout.

Furthermore, numerous studies have undertaken a comprehensive exploration of the relationship that exists between psychological capital and turnover intentions. These investigations have integrated various variables into the analysis, including leader–member exchange, emotional intelligence, organizational commitment, job satisfaction, and job performance [15,85,86]. Positive responses to workplace stimuli engender constructive attitudes, such as work engagement, empowerment, job satisfaction, and organizational commitment, which substantively contribute to personal and occupational well-being. [59,87,88]. This may elucidate the research community’s deep-seated interest in investigating factors that enhance individual psychological resources and mitigate burnout and turnover intentions [88,89].

Finally, for future research endeavors, it is advisable to consider the inclusion of additional variables alongside psychological capital and related constructs. This approach would facilitate a more comprehensive examination of their combined impact on occupational stress [13]. Building upon insights derived from previous studies, the following hypothesis is posited:

**Hypothesis 4 (H4).** 
*Burnout mediates the relationship between psychological capital and turnover intention.*


## 2. Materials and Methods

### 2.1. Study Design and Procedures

A cross-sectional study was conducted. An online survey was directed to healthcare professionals using a convenience sampling method in a total of six hospitals in the city of Guayaquil, Ecuador. The survey was first piloted with 58 healthcare professionals, including doctors and nursing staff. In this phase, some items reported as confusing or ambiguous by the participants were improved.

Hospitals’ department managers were contacted to request authorization and support to send the survey to the hospital’s healthcare personnel via e-mail, including only doctors and nurses. The e-mail indicated the objective and scope of the study. It was also stated that participation was voluntary. The survey began with brief instructions and an item asking the participant’s agreed informed consent. A total of 450 professionals received an e-mail with the online survey. The response rate was about 79%, with a total 356 surveys received.

### 2.2. Participants

The research exclusively studied doctors and nurses across various hospital departments as participants. Nursing assistants were not included since our focus is on healthcare professionals, meaning they must have a university degree. Physiotherapists, obstetricians, psychologists, dentists, and administrative personnel were also excluded from participation. Finally, medical staff above 65 years old were also excluded. Participants had a mean age of 27.18 years (SD = 7.73). The final sample, consisting of 320 healthcare professionals, was fairly balanced in terms of gender, with 52.6% being female and 47.3% male. In regard to educational attainment, 40.5% held post-graduate degrees, while 59.5% held only undergraduate degrees. The majority of the participants, accounting for 82.8%, were doctors, while the remaining 17.2% were nurses. As for workplace distribution, 61.1% of the professionals were affiliated with public hospitals, 22.1% worked in private hospitals, and 16.8% were employed by NGO healthcare institutions.

### 2.3. Research Instruments

A structured questionnaire was administered including the following constructs:

Burnout: The Burnout Inventory (MBI) by Maslach and Jackson [90] was applied. This study used the MBI version with 22 items grouped in three dimensions. The first dimension, Exhaustion, was measured with nine items. The second dimension, Depersonalization, consisted of five items, and the last dimension, Personal accomplishment, was composed of eight items. Items were rated using a 7-point Likert scale, ranging from 1 (never) to 7 (everyday). The overall reliability index for the scale in this study was acceptable (ω = 0.695). The reliability indices are reported in Section 4.

Psychological capital: This construct was measured with 12 items proposed by Luthans and colleagues [9]. Items were rated using a 6-point Likert scale ranging from 1 (strongly disagree) to 6 (strongly agree). The construct was composed of four dimensions. The first dimension included self-efficacy, resilience, optimism, and hope. The overall reliability of this scale was satisfactory (ω = 0.868).

Turnover intention: Items designed by Roodt and validated by Bothma and Roodt [91] were used to measure turnover intention among healthcare professionals. Items were rated using a 5-point Likert scale ranging from 1 (never) to 5 (everyday). The overall reliability of this scale was satisfactory (ω = 0.843).

### 2.4. Data Analysis

The normality of the data was assessed by inspecting the skewness and kurtosis. Values outside the interval [−2, +2] and [−7, +7] for skewness and kurtosis, respectively, were considered a sign of non-normality of the data [92]. The Mahalanobis-squared distance was used to detect outliers, and observations with *p*-values below 0.001 [93] were eliminated from the dataset. These calculations were carried out using IBM SPSS v.25.

Next, the constructs’ reliability was assessed using McDonald’s omega (ω), which is more appropriate than Cronbach’s alpha since this last reliability measure is not SEM based and because it usually underestimates reliability and requires the assumption of tau-equivalence, which is not justified for latent constructs [94,95]. Acceptable values for McDonald’s omega are 0.70 or higher [96].

Confirmatory factor analysis (CFA) was conducted to evaluate the validity of each construct. CFA models were estimated using a robust maximum likelihood (MLR) estimator to deal with non-normality of the data. The model’s goodness-of-fit was assessed using common indices including the comparative fit index (CFI), the goodness-of-fit index (GFI), the normed fit index (NFI), the Tucker–Lewis index (TLI), the root mean square error of approximation (RMSEA), and the standardized root mean square residual (SRMR). Factor loadings were examined and items with values above 0.50 were retained to ensure convergent validity. The average variance extracted (AVE) was also calculated, and values above 0.50 were considered evidence supporting convergent validity [94].

Finally, the hypotheses were evaluated through the structural model, focusing on turnover intention as the outcome variable, with psychological capital and burnout serving as the predictor variables. The mediation analysis was conducted using a bootstrapping process based on 1000 iterations and a 95% confidence interval. CFA and SEM models were estimated using the Lavaan package in R-Studio.

## 3. Results

### 3.1. Correlational Analysis

Correlational analyses, along with the mean and standard deviation of each construct, are presented in Table 1. Factor scores were computed as the average among the items in each construct. These calculations resulted in scores where higher values represented higher levels of psychological capital, burnout, or turnover intention. According to these scores, the participants showed a high level of psychological capital (M = 4.77, SD = 1.02), with higher levels in hope (M = 4.83, SD = 1.11) and optimism (M = 4.81, SD = 1.28) dimensions. In contrast, healthcare professionals showed low levels of burnout (M = 0.36, SD = 0.99), explained by low levels of depersonalization (M = 2.56, SD = 1.25) and high levels of personal accomplishment (M = 5.56, SD = 1.09). Finally, scores on turnover intention were moderate (M = 2.89, SD = 1.03).

The results of the correlation analysis showed that psychological capital and its dimension were negatively and significantly associated with burnout and turnover intention, as expected, except for personal accomplishment, which was composed of inverse items in the burnout scale. In contrast, there was a positive correlation among the burnout constructs and turnover intention. The strongest correlation was found between psychological capital and personal accomplishment (r = 0.616, *p* < 0.01). Furthermore, overall burnout and turnover intention correlation were stronger (r = 0.582, *p* < 0.01) than overall psychological capital and turnover intention (r = −0.328, *p* < 0.01).

### 3.2. Reliability and Confirmatory Analysis

CFA was used to confirm the construct dimensionality. In the estimated model, psychological capital and burnout were modelled as second-order constructs and turnover intention was modelled as a unidimensional construct. This specification showed acceptable goodness-of-fit indices (CFI = 0.914; NFI = 0.899; TLI = 0.905; RMSEA = 0.054 [90% CI: 0.049–0.059]; SRMR = 0.089.) Table 2 reports factor loadings, CR, and AVE values for each construct. All items in psychological capital were higher than the 0.50 cut-off value. In the case of burnout, four items, including de15, ee06, ee16, and pa04, were dropped since their factor loadings were below 0.30 and the construct’s reliability and convergent validity improved when omitting these items. Furthermore, items ti05 and ti06, with factor loadings below 0.45, were also dropped to achieve convergent validity and high reliability in the turnover intention construct.

AVE was above the recommended 0.50 cut-off value [94] in all cases except the dimensions depersonalization (AVE = 0.457) and resilience (AVE = 0.490). However, values ranging from 0.40 to 0.49 are acceptable when reliability levels are adequate [97]. These results support the validity and reliability of the constructs included in the model.

### 3.3. Structural Modeling and Hypotheses Testing

The research model was estimated using SEM. The estimations indicated that psychological capital had a significant negative direct effect on burnout (β = −0.490, *p* < 0.001) and there was also a significant positive direct effect of burnout on turnover intention (β = 0.803 *p* < 0.001). These results are supporting evidence for hypotheses H2 and H3. In contrast, psychological capital did not have a significant direct effect on turnover intention (β = 0.001, n.s.). Thus, hypothesis H1 was not confirmed.

Regarding the mediation of burnout, estimations and the bootstrap method showed a significant indirect effect of psychological capital on turnover intention through burnout (β = −0.393, CI: LL = −0.587, UL = −0.241, *p* < 0.001), supporting H4. The total effect of psychological capital on turnover intention was also significant (β = −0.394, CI: LL = −0.530, UL = −0.234, *p* < 0.001). The results of this model are presented in Table 3 and Figure 1. The goodness-of-fit indexes for this model were acceptable (CFI = 0.923; NFI = 0.899; TLI = 0.914; RMSEA = 0.053 [90% CI: 0.047–0.058]; SRMR = 0.088).

## 4. Discussion

The results of this study elucidate the nuanced interrelationships among psychological capital, burnout, and turnover intention within the healthcare sector, offering critical insights with manifold implications. The participants demonstrated a notable level of psychological capital, particularly in hope and optimism, dimensions known for their stress-buffering effects, which is consistent with the findings of other scholars [9,10,15].

The inverse relationship between psychological capital and burnout aligns with the conservation of resources theory, where individuals with abundant psychological resources are less susceptible to the deleterious effects of occupational stressors [98]. Interestingly, while psychological capital did not directly influence turnover intention, its impact was significantly mediated by burnout levels, indicating that psychological strengths may retain employees by reducing burnout. These findings are comparable to the study carried out by other authors who measured the mediating effect of occupational stress in the relationship between psychological capital and turnover intention [13].

The pronounced negative correlation between psychological capital and burnout, particularly the strong association with personal accomplishment, underscores the protective role of positive psychological resources. This suggests that by fostering a work environment that enhances psychological capital, healthcare organizations can mitigate the emotional exhaustion and depersonalization components of burnout, subsequently bolstering feelings of professional efficacy. These findings echo previous research studies advocating for the development of psychological capital as a strategy for burnout prevention [11,42,43,77,78,79,80,81].

The positive correlation between burnout and turnover intention is congruent with the broader literature, signifying that burnout remains a critical precursor to employees’ decisions to leave their jobs [22,23,65,69].

The CFA and subsequent item analysis uphold the structural integrity of the constructs used, even though certain items were excised to enhance the model’s validity and reliability. The slight underperformance of the depersonalization and resilience dimensions in the AVE metric, although within acceptable bounds, suggests potential areas for further refinement and research.

In hypothesis testing via bootstrapping in SEM, psychological capital’s lack of a direct effect on turnover intention, yet substantial indirect effect through the mediation of burnout, is revelatory. It highlights burnout’s role as a mediator, a finding that adds complexity to our understanding of how psychological resources influence employee satisfaction and work outcomes. This mediated relationship suggests that interventions to reduce turnover should perhaps target burnout reduction through the enhancement of psychological resources, rather than by addressing turnover intention directly. This indirect strategy may be more efficacious given the intricate interplay of the constructs involved [99].

The study’s findings hold substantial implications for healthcare management and policy. Given the established links among psychological capital, burnout, and turnover intention, healthcare administrators are urged to devise and implement strategies that bolster healthcare professionals’ psychological capital.

By enhancing attributes such as hope, resilience, and optimism, not only can the well-being of healthcare workers be improved, but institutions may also observe a decrease in turnover intentions, mediated by reduced levels of burnout. Furthermore, the results advocate for a deeper approach to addressing the staffing crisis in healthcare, suggesting that efforts to retain staff must operate at multiple levels of the individual experience, rather than focusing solely on conventional job satisfaction or organizational commitment metrics.

### 4.1. Implications

The findings of this study, examining the relationships between psychological capital, burnout, and turnover intention among healthcare professionals, have significant implications for healthcare management, public health policy, and workplace mental health strategies. These implications are particularly relevant given the critical role healthcare professionals play in society.

In the realm of healthcare management, the necessity to prevent burnout is paramount. As burnout directly impacts the health and performance of healthcare professionals, it consequently affects the quality of patient care and healthcare system efficiency. Public health initiatives should therefore focus on creating and implementing programs that identify and address the factors contributing to burnout in healthcare settings.

The strong positive impact of psychological capital on reducing burnout suggests that enhancing these attributes in healthcare professionals is crucial. Rather than viewing psychological capital as an innate characteristic, healthcare institutions should treat it as a skill that can be developed through continuous professional development. This can include resilience training, mindfulness and stress management workshops, and emotional intelligence training integrated into the professional development programs of healthcare workers.

Further, healthcare organizations are encouraged to foster a work culture that nurtures psychological capital. This involves creating supportive and positive work environments, providing opportunities for career advancement, recognizing and rewarding achievements, and ensuring a healthy work–life balance. Such an organizational culture not only enhances the mental well-being of healthcare workers but also indirectly reduces their turnover intentions by mitigating burnout.

Integrating psychological capital into existing mental health programs within healthcare settings can further enhance their effectiveness. These programs should include proactive strategies to develop psychological resilience and strength, thereby contributing to the overall mental health and well-being of healthcare professionals.

Collaboration between healthcare organizations, public health authorities, and mental health professionals is essential to develop and implement comprehensive mental health strategies. These strategies should be aimed at developing psychological capital among healthcare professionals and addressing burnout, thereby improving their mental health and job satisfaction.

In summary, this study highlights the need for a holistic approach in healthcare settings, focusing on enhancing psychological capital among healthcare professionals to reduce burnout and turnover intention. By doing so, healthcare organizations can not only improve the well-being of their workforce but also ensure the delivery of high-quality patient care and the effective functioning of the healthcare system. This approach is a crucial step towards a more comprehensive understanding and enhancement of health and well-being in the healthcare profession.

### 4.2. Limitations

One notable limitation of the study is the methodological approach, primarily the reliance on self-reported data, which raises concerns regarding the potential for common method bias. Participants’ responses may be influenced by a range of subjective factors, including mood, current job satisfaction, or by social desirability bias, thereby not accurately reflecting the objective reality of the constructs measured, such as psychological capital, burnout, or turnover intention. The cross-sectional nature of the study further compounds this issue as it captures only a snapshot of participants’ experiences and perceptions at a single point in time, which may not accurately represent these variables’ fluctuating nature over time. This design precludes establishing causality among the constructs, making it challenging to deduce definitive relationships between psychological capital, burnout, and turnover intention.

Another limitation is the potential oversimplification of the constructs involved, particularly psychological capital and burnout. These constructs are multifaceted and influenced by an interplay of individual, organizational, and possibly cultural factors, which the study might not have fully captured or considered. For instance, the dimensionality of psychological capital could be influenced by factors beyond hope, resilience, optimism, and efficacy, such as emotional intelligence or cultural values. Similarly, burnout in healthcare professionals might be affected by work–life balance, quantitative workload, organizational culture, leadership style, or even patient-related stressors, which are aspects not evidently accounted for in the study. Additionally, the exclusion of certain items from the burnout and turnover intention constructs to improve model fit indicates potential issues with the measurement instruments’ validity and reliability in capturing the full scope of these complex constructs. These limitations suggest a need for caution in generalizing the study’s findings without more comprehensive, longitudinal, and culturally nuanced research.

Finally, future research should aim to understand how individual resilience interacts with organizational culture, leadership styles, and policy frameworks and also further explore what factors could predict burnout. Furthermore, longitudinal studies are recommended to ascertain causality and observe the evolution of psychological capital and its impact on burnout and turnover intentions over time.

## 5. Conclusions

The aim of this study was to analyze the mediating role of burnout in the relationship between psychological capital and turnover intention among healthcare professionals. The findings provide insightful contributions to the existing literature in this field. Firstly, the high levels of psychological capital observed in participants, particularly in the domains of hope and optimism, are noteworthy. This is complemented by the low levels of burnout and moderate levels of turnover intention, suggesting a potentially protective role of psychological capital in mitigating burnout and its subsequent impact on turnover intentions.

The correlational analysis revealed a significant negative association between psychological capital and both burnout and turnover intention, except for the personal accomplishment component of burnout. This highlights the complex dynamics between these constructs and underscores the importance of psychological capital in the professional well-being of healthcare workers.

Confirmatory factor analysis and reliability testing further validated the constructs used in the study, affirming the robustness of the findings. Importantly, the structural modeling and hypothesis testing revealed a significant negative direct effect of psychological capital on burnout and a significant positive direct effect of burnout on turnover intention. However, psychological capital did not have a direct significant effect on turnover intention, indicating that its influence is primarily exerted through its impact on burnout.

The mediation analysis showed a significant indirect effect of psychological capital on turnover intention via burnout, substantiating the hypothesis that psychological capital can buffer the adverse effects of burnout, thereby indirectly reducing turnover intentions. This indicates a crucial pathway through which psychological capital contributes to workforce stability in healthcare settings.

Finally, the study underscores the critical role of psychological capital in reducing burnout and indirectly influencing turnover intentions among healthcare professionals. These findings suggest that enhancing psychological capital could be a valuable strategy in managing workforce dynamics, particularly in high-stress environments like healthcare. This approach could lead to improved employee well-being and reduced turnover, which are essential for the delivery of high-quality healthcare services.

## Figures and Tables

**Figure 1 ijerph-21-00185-f001:**
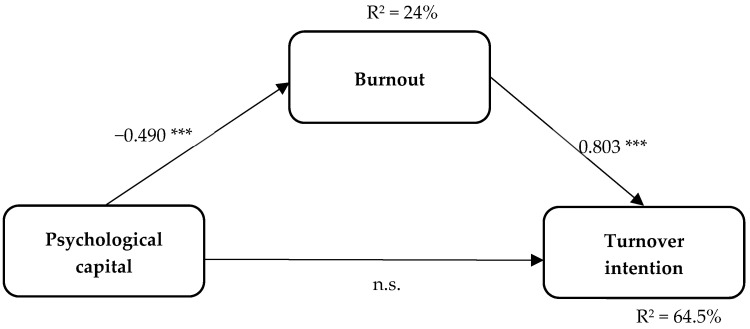
SEM results among psychological capital, burnout, and turnover intention. Note: n.s.: not significant. *** *p* < 0.001.

**Table 1 ijerph-21-00185-t001:** Mean, standard deviation, and correlation coefficient between constructs.

Variable	Mean ± SD	(1)	(2)	(3)	(4)	(5)	(6)	(7)	(8)	(9)
(1) Psychological capital	4.77 ± 1.02									
(2) Hope	4.83 ± 1.11	0.856 **								
(3) Optimism	4.81 ± 1.28	0.851**	0.693 **							
(4) Resilience	4.77 ± 1.12	0.779 **	0.604 **	0.602 **						
(5) Self-efficacy	4.69 ± 1.29	0.825 **	0.641 **	0.570 **	0.523 **					
(6) Burnout	0.36 ± 0.99	−0.593 **	−0.539 **	−0.522 **	−0.406 **	−0.485 **				
(7) Emotional exhaustion	4.07 ± 1.59	−0.447 **	−0.399 **	−0.386 **	−0.278 **	−0.355 **	0.848 **			
(8) Depersonalization	2.56 ± 1.25	−0.392 **	−0.370 **	−0.326 **	−0.243 **	−0.387 **	0.817 **	0.588 **		
(9) Personal accomplishment	5.56 ± 1.09	0.616 **	0.573 **	0.569 **	0.508 **	0.449 **	−0.590 **	−0.248 **	−0.311**	
(10) Turnover intention	2.89 ± 1.03	−0.328 **	−0.355 **	−0.270 **	−0.153 **	−0.222 **	0.582 **	0.662 **	0.425 **	−0.221 **

Notes. SD = Standard deviation. Correlations were obtained using Spearman rank correlation. Significance level: ** *p* < 0.01.

**Table 2 ijerph-21-00185-t002:** Confirmatory factor analysis and reliability results.

Item	Statement	Loadings	CR	AVE
Psychological capital			0.868	0.805
Self-efficacy		0.819	0.778	0.606
se01	Confidence representing work area in meetings with management	0.683		
se02	Confidence contributing to discussions about the organization’s strategy	0.773		
se03	Confidence presenting information to a group of colleagues	0.897		
Hope		0.972	0.872	0.618
ho04	In a jam, I can think about many ways to get out of it	0.752		
ho05	See myself as being successful at work	0.724		
ho06	Can think about many ways to reach my current work goals	0.854		
ho07	Meeting the work goals	0.808		
Resilience		0.844	0.748	0.490
re08	Can be ‘on my own’, so to speak, at work if I have to	0.653		
re09	Take stressful things at work in stride	0.695		
re10	Can get through difficult times at work, I have experienced difficulty	0.756		
Optimism		0.904	0.813	0.684
op11	Look on the bright side of things regarding my job	0.801		
op12	Optimistic about what will happen in the future as it pertains to work	0.854		
Burnout			0.695	0.535
Emotional exhaustion		0.826	0.927	0.598
ee01	Emotionally drained from my work	0.856		
ee02	Used up at the end of the workday	0.831		
ee03	Fatigued when I get up in the morning	0.864		
ee06	Working with people all day is really a strain for me	0.649		
ee08	Burned out from my work	0.904		
ee13	Frustrated by my job	0.750		
ee14	Working too hard	0.765		
ee16	Working with people directly puts too much stress	0.588		
ee20	At the end of my rope	0.741		
Depersonalization		0.896	0.694	0.457
de05	Treat some people as if they were impersonal “objects”	0.516		
de10	Callous toward people, as I took this job	0.676		
de11	Worry that this job is hardening me unemotionally	0.730		
Personal accomplishment		0.350	0.806	0.592
pa12	Energetic	0.514		
pa17	Create relaxed atmosphere with others	0.702		
pa18	Exhilarated after working closely with people	0.885		
pa19	Accomplished worthwhile things in this job	0.655		
Turnover intention			0.843	0.536
ti01	Considering leaving the job	0.766		
ti02	Satisfying job, fulfilling personal needs	0.665		
ti03	Frustrated when not given the opportunity to achieve personal goals	0.716		
ti04	Dreaming about getting another job that will better suit personal needs	0.746		

Notes. CR: Composite reliability based on McDonald Omega, AVE: Average variance extracted. All factor loadings were significant at *p* < 0.001.

**Table 3 ijerph-21-00185-t003:** Structural model estimation results.

Effects	β	SE	Boot 95% CI	
			LL	UL
Direct effects				
Psychological capital -> Burnout	−0.490 ***	0.079	−0.637	−0.324
Psychological capital -> Turnover intention	0.001	0.087	−0.160	0.182
Burnout -> Turnover intention	0.803 ***	0.064	0.680	0.939
Indirect effect				
Psychological capital -> Burnout -> Turnover intention	−0.393 **	0.081	−0.587	−0.241
Total indirect effect	−0.393 **	0.079	−0.530	−0.234

Notes. β: standardized coefficients, SE: standard error, LL: lower limit, UL: upper limit. In total, 95% CI calculated using bootstrapping methods on 1000 iterations. *** *p* < 0.001, ** *p* < 0.01.

## Data Availability

The data presented in this study are available on request to the corresponding authors. The data are not publicly available due to privacy concerns.

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
