# Peer review of "Psychological Capital and Turnover Intention: The Mediating Role of Burnout among Healthcare Professionals"

_ijerph, 2024, doi:10.3390/ijerph21020185_

Round 1
Reviewer 1 Report
Comments and Suggestions for Authors
Comment
This study examined the mediation effect of burnout on the relationship between psychological capital and turnover intention among healthcare professionals. This manuscript may contribute to this area of research. Further attention to the issues presented below would strengthen the manuscript.
#1
Regarding the relationship among psychological capital, burnout, and turnover intention. The author examines a model in which burnout mediates the relationship between psychological capital and turnover intention.
This is just my opinion. A model in which psychological capital moderates the association between burnout and turnover intention would be more natural, and the applicability of the findings would be higher. I believe that working under high-stress conditions that cause burnout is more likely to produce turnover intention, and that this effect may be buffered in those with high psychological capital.
I would like to ask the author's opinion on this issue.
Author Response
Dear Reviewer,
We are delighted to submit the revised version of our original article, titled "Psychological Capital and Turnover Intention: The Mediating Role of Burnout among Healthcare Professionals" following the incorporation of the valuable suggestions provided by the reviewers.
In the file attached, you will find our detailed response to your comments and suggestions.
Sincerely,
Laura Zambrano
Rubén Guevara

Reviewer 2 Report
Comments and Suggestions for Authors
Thank you for inviting me to review this manuscript, which reports on a study regarding the role of burnout, psychological capital and turnover intentions among healthcare professionals. Overall, I do not have many issues with this study.
The distinction between the sections ‘Introduction’ and ‘Literature Review’ is not clear. It seems that the literature review addresses the same topics as the introduction, but they are serving a different purpose. The introduction ends with the formulation of the aim of the study (as it should be), and the hypotheses are formulated in the ‘review’ section.
Would it make sense to integrate both sections into one?
Materials and Methods
Please report the inclusion and exclusion criteria.
Conclusions
I would limit this section to conclusions without repeating study limitations.
I hope these few comments may help in revising the manuscript. Good luck.
Comments on the Quality of English LanguageMinor corrections.
Author Response

(The authors gave the same response as above.)

Round 2
Reviewer 1 Report
Comments and Suggestions for Authors
I have confirmed that the author has addressed my concerns successfully in the revised version of the manuscript.